

# Opposite Effects of Aerosols on Daytime Urban Heat Island Intensity between Summer and Winter

Wenchao Han[1,2], Zhanqing Li[1,2*], Fang Wu[1,2], Yuwei Zhang[3], Jianping Guo[4], Tianning Su[2], Maureen Cribb[2], Tianmeng Chen[4], Jing Wei[1,2], Seoung-Soo Lee[5]

1. State Key Laboratory of Remote Sensing Science, College of Global Change and Earth System Science, Beijing Normal University, Beijing 100875, China
2. Department of Atmospheric and Oceanic Science and Earth System Science Interdisciplinary Center, University of Maryland, College Park, Maryland, 20740, USA
3. Atmospheric Sciences and Global Change Division, Pacific Northwest National Laboratory, Richland, Washington, 99352, USA
4. State Key Laboratory of Severe Weather, Chinese Academy of Meteorological Sciences, Beijing 100081, China
5. San Jose State University Research Foundation, San Jose, California, 95192, USA

Correspondence to: Zhanqing Li (zli@atmos.umd.edu)

**Abstract**

The urban heat island intensity (UHII) is the temperature difference between urban areas and their rural surroundings. It is commonly attributed to changes in the underlying surface structure caused by urbanization. Air pollution caused by aerosol particles can affect the UHII by changing the surface energy balance and atmospheric thermodynamic structure. By analyzing satellite data and ground-based observations collected from 2001 to 2010 at 35 cities in China and using the WRF-Chem model, we found that aerosols have very different effects on daytime UHII in different seasons: reducing the UHII in summer, but increasing the UHII in winter. The seasonal contrast in the spatial distribution of aerosols between the urban centers and the suburbs lead to a spatial discrepancy in aerosol radiative effect (SD-ARE). Additionally, different stability of the planetary boundary layer induced by aerosol is closely associated with a dynamic effect (DE) on the UHII. SD-ARE reduces the amount of radiation reaching



the ground and changes the vertical temperature gradient, whereas DE increases the stability of the planetary boundary layer and weakens heat release and exchange between the surface and the PBL. Both effects exist under polluted conditions, but their relative roles are opposite between the two seasons. It is the joint effects of the SD-ARE and the DE that drive the UHII to behave differently in different seasons, which is confirmed by model simulations. In summer, the UHII is mainly affected by the SD-ARE, and

the DE is weak, and the opposite is the case in winter. This finding sheds a new light on the impact of the interaction between urbanization-induced surface changes and air pollution on urban climate.

## 1. Introduction

The global population has been increasingly concentrated in cities (Heilig 2012). Urbanization in China

has dramatically increased from 26% in 1990 to 60% in 2018, resulting in a marked change of landscape. It has a significant impact on the urban and rural climate and will continue to make an impact as cities continue to develop (Han et al. 2014).

Urbanization leads to a dramatic change in the underlying surface structure, properties, and spatial distribution of a city, such as a reduction in green areas and a corresponding increase in impervious areas.

These changes increase the temperature difference between urban and rural areas, which is known as the urban heat island (UHI) intensity (UHII) (e.g., Kalnay and Cai 2003, Zhao et al. 2014, Zhao et al. 2016, Zhou et al. 2016, Yang et al. 2017). The UHI also affects the structure and movement of cloud systems (Changnon and Westcott 2002, Kug and Ahn 2013, Pinto et al. 2013). The diurnally and seasonally varying UHI is affected by many factors, such as weather and climatic regimes, urban impervious surfaces,

anthropogenic heat, air pollution, and urban 3D structure (Oke 1982, Morris and Simmonds 2000, Kim and Baik 2002, Gedzelman et al. 2003, Ryu and Baik 2012, Ding et al. 2016, Yang Y et al. 2019). It is well established that cities are the largest sources of anthropogenic heat emissions as by-products from industrial and human activities. Human activities can also generate large amounts of aerosols that can reduce air quality, change the physical and chemical properties of the atmosphere, and endanger human

health (Sanap and Pandithurai 2015, Cohen et al. 2017, Wei et al. 2019a, b). With increasing urbanization in the future, cities are likely to influence local and regional weather and climate to greater degrees.





Aerosols can also alter the radiation balance of the climate system. Their thermodynamic effect reduces the amount of radiation reaching the ground, and their microphysical effect can influence cloud properties and precipitation regimes through their impacts on cloud microphysical and dynamic processes

(Rosenfeld et al. 2008, Li et al. 2011, Fan et al. 2013, Li et al. 2016, Guo et al. 2018, Liu et al. 2019). The effect of urbanization on clouds and precipitation has been the focus of many studies (Changnon et al. 1977, Ackerman et al. 1978, Changnon et al. 1991, Shepherd et al. 2002, Shepherd and Burian 2003). Aerosols can increase cloudiness and cloud thickness and thus change the stability of the planetary boundary layer. In humid regions, aerosols may reduce the frequency of light rain but increase heavy

rainfall, while in dry areas, aerosols aggravate droughts. Aerosols can also intensify convection by delaying the occurrence of convection and enhancing gust fronts (Khain et al. 2005, Carrió et al. 2010, Carrió and Cotton 2011, Wang et al. 2011, Han et al. 2012, Lee and Feingold 2013, Li et al. 2017, Guo et al., 2016a).

UHI, surface roughness and higher aerosol concentrations have been proposed to explain observed urban

clouds and precipitation anomalies. Increased urban surface roughness likely does not play a major role in urban-induced precipitation. Rather, the UHI and higher aerosol concentrations may play more important roles (Han et al. 2014). The UHI can alter the water vapor flux (accelerate evaporation), reduce horizontal wind speeds and enhance vertical turbulence, reduce the temperature difference between daytime and nighttime, increase the absorption rate of solar radiation by land, and change underlying

surface characteristics (e.g., sensible heat dissipation, convection efficiency, evaporation and cooling, sunlight reflection, and anthropogenic heat transfer) (Jauregui and Romales 1996, Taha 1997, Bornstein and Lin 2000, Givati and Rosenfeld 2004, Grimmond 2007, Carrió et al. 2010, Zhao et al. 2014, Kaspersen et al. 2015, Yang B et al. 2019).

The UHI and aerosols may interact over cities. Aerosols generally reflect and absorb solar radiation and

reduce the amount of shortwave radiation reaching the ground, i.e., the cooling effect of aerosols on ground temperature. Some numerical modelling studies have demonstrated that landscape change reduces near-surface concentrations of $PM_{2.5}$ and that the UHI effect can influence the dispersion of air pollutants (Liu et al. 2009, Liao et al. 2015, Tao et al. 2015, Zhong et al. 2017, 2018). Moreover, aerosols can enhance the UHI at night (by 0.7±0.3 K) for semi-arid cities, and the UHI alters the aerosol concentration



(Cao et al. 2016, Fallmann et al. 2016, Lai 2016). Heavy pollution can reduce UHII in China, especially during the day (Wu et al. 2017, Yang et al. 2020).

Weather Research and Forecasting/Chemistry (WRF-Chem) model are used extensively in the simulation and prediction of air quality, the aerosol radiation effect and aerosol-cloud interactions, and the change of meteorological fields and regional climate (Grell et al. 2005, Chapman et al. 2009). Coupled with the

urban canopy model, WRF can account for the influences of aerosols and land surface changes on the radiative processes if such parameters are fed to the model as aerosol loading and single scattering albedo, surface albedo and thermal emissivity, roughness, etc (Miao et al. 2009, Chen et al. 2011). Many previous pertinent studies are done to date just focused on the annual effects without investigating any seasonal differences and the underlying mechanism. This study aims to fill this gap by analyzing the annual and

seasonal effects of aerosols on UHII and proposing mechanisms that may explain the seasonal differences.

## 2. Methods

### 2.1 Study areas and data

Thirty-five big cities evenly distributed across China were selected in our study. Table S1 lists these cities

of different sizes. They represent the major and well-developed metropolitan regions in China. The population and urban areas of these cities have increased faster and/or more dramatic than those of other cities from 2001 to 2015.

Data used in this study include Land Satellite Thematic Mapper/ Enhanced Thematic Mapper (Landsat TM/ETM+) and Moderate Resolution Imaging Spectroradiometer (MODIS) products [including land

surface temperature (LST) and aerosol optical depth (AOD)], ground-based data from meteorological stations, particulate matter ($PM_{2.5}$) concentrations, and sounding data.

Landsat data are used to identify and outline urban areas and urban contour. The spatial resolution is 30 m. Summertime (June, July, and August) images from 2000 and 2015 were examined to ensure the accuracy and consistency of results.

The MODIS LST product (MYD11A2) at a 1-km spatial resolution was used to calculate urban and rural UHIIs. Since this study is mainly focused on the daytime UHI effect, only data (eight-day clear-sky LST observations with 1 km spatial resolution) at 13:30 BJT for the period 2001–2015 were used. The





MYD11A2 product uses the MODIS cloud mask product (MYD35) to filter out cloudy conditions. A generalized split-window algorithm is applied using MODIS data in two longwave bands in the atmospheric window to correct for atmospheric water vapor, haze effects, and the sensitivity to errors in the surface emissivity. To obtain the LST from brightness temperatures, changes in surface emissivity have been accounted for (Wan and Dozier 1996, Snyder et al. 1998, Wang and Liang 2009, Yu et al. 2011, Cao et al. 2016).

The MODIS Multi-Angle Implementation of Atmospheric Correction (MAIAC) AOD product is used that has a 1-km spatial resolution with daily global coverage. It was retrieved by virtue of a time series analysis and a combination of pixel- and image-based processing to improve the accuracies of cloud detection, aerosol retrievals, and atmospheric correction (Lyapustin et al. 2011a, 2011b, 2012).

A large volume of meteorological data are analyzed including visibility, surface wind speed, temperature, precipitation, and other parameters every three hours, together with hourly $PM_{2.5}$ data in urban and surrounding rural areas. To be consistent with the satellite imaging time (13:30 BJT), the meteorological data and $PM_{2.5}$ data observed at 13:00 and 14:00 BJT are selected. Due to the lack of long-term records of aerosol concentration, visibility is frequently used as a proxy for aerosol loading (Wang et al. 2009, Wu et al. 2012, Yang et al. 2013).

The L-band sounding data are employed that were acquired at the five radiosonde stations in Beijing, Chengdu, Nanjing, Shenyang, and Xi'an operated by the China Meteorological Administration since 2006. They contain the high-resolution profiles of temperature, pressure, relative humidity, and wind speed and direction at 08:00 Beijing time (BJT, UTC+8) and 20:00 BJT (Zhang et al., 2018; Lou et al., 2019). The data quality of radiosonde measurements has been well validated and is good enough to study the UHI effect (Guo et al., 2016b).

## 2.2 Extracting urban impervious surfaces and urban contours

Indices commonly used to extract built-up areas include the Difference Built-up Index (DBI), the Index-based Built-Up Index (IBI), and the Normalized Difference Built-up Index (NDBI). Another index, the Soil-adjusted Vegetation Index (SAVI), is a modification of the normalized difference vegetation index



that corrects for the influence of *soil* brightness when the vegetative cover is low (Huete 1988, Qi et al. 1994, Rondeaux et al. 1996). After some tests, the difference $NDBI - SAVI$ were used to extract urban impervious surfaces because of its ability to differentiate urban impervious surfaces from other land-use types:

$$NDBI = \frac{\rho_5 - \rho_4}{\rho_5 + \rho_4}, \qquad\qquad (1)$$

$$SAVI = \frac{\rho_4 - \rho_3}{\rho_4 + \rho_2 + L}(1 + L), \qquad\qquad (2)$$

where $L$ is the soil adjustment factor whose value is 0.5, and $\rho_n$ is the Landsat reflectance of band $n$. We then used different thresholds to extract urban impervious surfaces after calculating $NDBI - SAVI$. Results are verified by the Google Earth and a land-use map with a 1:100,000 scale from the Resource and Environment Science Data Center of the Chinese Academy of Sciences.


**2.3 Research windows**

Many previous studies have extracted urban areas from nighttime stable-light data. However, the spatial resolution of such data is low, so the extraction accuracy would be significantly affected in urban areas with uneven zoning and in regions with irregular urban development as in most municipalities in China.

The TM/ETM+ data are used to accurately extract the physical boundaries of urban areas. The difference in the underlying surfaces of urban and rural areas forms the basis of the urban physical boundary extraction. Urban surfaces are generally covered by impervious materials, and rural surfaces are mainly covered by natural surfaces. The influence of the UHI is not only felt within the physical boundaries of urban areas but also beyond it. In terms of area, this influence can extend to 2–4 times the extent of an

urban area. In terms of distance, the influence of the UHI can be felt as far as 3–6 km away from an urban physical boundary (Zhou et al. 2015).

For each city, nine research windows (6 km x 6 km each) were selected. The windows include one urban window, four suburban windows, and four rural windows. For the study period considered (2001–2015), the urban window represents an area that remained urban and developed during this time. The suburban

windows represent areas that were vegetated before the study period. As cities expand, these areas were gradually replaced by urban impervious surfaces from 2001 to 2015. The rural windows represent areas





that remained vegetated during the study period. These windows were 10 km away from the urban physical boundary to ensure that these windows were not or weakly affected by the UHI. The elevations of the areas covered by each window are within 100 m of each other for a given city based on DEM 170 (Digital Elevation Model) data. Water bodies are excluded. Figure S1 shows the spatial distribution of the nine research windows for a given city. The UHII is the temperature difference between the average temperature of the urban core window and the average temperature of rural windows.

## 2.4 Aerosol parameters (AOD, $PM_{2.5}$)

Validation using Aerosol Robotic Network AOD retrievals shows that the MAIAC and MODIS aerosol retrieval algorithms have similar accuracies over dark and vegetated surfaces and that the MAIAC algorithm generally improves the accuracies of AOD retrievals over bright surfaces such as deserts and urban surfaces (Lyapustin, et al. 2011a, 2011b, 2012, Wei et al., 2019c, Zhang et al. 2019). Sounding data and $PM_{2.5}$ measurements were available from 2013 to 2015. MAIAC AOD retrievals for each area were 180 averaged to obtain the spatial distribution of AOD over each city, then the difference of AOD between urban and rural areas was calculated.

## 2.5 WRF-Chem model simulation

The model used in this study is WRF-Chem 3.9.1 coupled with a single-layer urban canopy model. As 185 shown in Figure S6, the domain with a horizontal grid resolution of 3 km and 50 vertical levels from the surface to 50 hPa is used. To better characterize the planetary boundary layer, 16 layers are set below 1km, of which the first layer is about 47m in Beijing. Meteorological fields are provided by the National Centers for Environmental Prediction Final Analysis data (NCEP-FNL) with 6-h temporal frequency and $1°\times1°$ spatial resolution. The chemical lateral boundary and initial conditions were provided by GEOS-190 CHEM model. The land cover is derived from the IGBP-Modified MODIS 20-category Land Use Categories dataset. Monthly $0.25°\times0.25°$ anthropogenic emissions of aerosols and precursors are obtained from the Multi-resolution Emission Inventory for China (MEIC, 2012) (http://www.meicmodel.org), which provide monthly mean emission data of $SO_2$, $NO_x$, CO, NMVOC, $NH_3$, BC, OC, $PM_{2.5}$, $PM_{10}$, and $CO_2$. The biogenic emission data is provided by the Model of Emissions of Gases and Aerosols from





Nature (MEGAN) (Guenther et al., 2006. Sakulyanontvittaya et al., 2008). The Fire Inventory from NCAR (FINN) model provides the biomass burning emission data (Wiedinmyer et al., 2011). Carbon-Bond Mechanism version Z (CBMZ) chemical mechanism and Model for Simulating Aerosol Interactions and Chemistry (MOSAIC) are used in this simulation (Zaveri and Peters 1999, Zaveri et al. 2008). Other details of schemes used in simulations are shown in Table S3. The simulations are initiated at 1200 UTC

June 2015 for summer and 1200 UTC 01 January 2015 for winter. The meteorological fields are reinitialized every 48 hours. We conducted four sets of model experiments (Table S2) to investigate the aerosol radiative impact for both summer and winter (a) A1Summer with aerosol radiative effect turned on, (b) A0Summer with aerosol radiative effect turned off, (c) A1Winter with aerosol radiative effect turned on, and (d) A0Winter with aerosol radiative effect turned off. To be consistent with the observation

analysis, we select clear-day simulations as the analysis time period by excluding the first 3-day simulation for chemistry spin-up (Table S2).

## 3. The Urban Heat Island (UHI) effect

We used the difference NDBI − SAVI to extract urban impervious surfaces, and then determined urban

contours based on the identification of impervious surfaces. Figure 1 shows the urban contours of all cities.





**Figure 1.** Main urban contours of the 35 cities. Blue contours outline urban boundaries in 2001 and red contours urban boundaries in 2015. The surface height (in meters above sea level) is indicated by color
shading.

Figure S2 shows UHII and visibility trends. For most cities, no matter before or after 2008, there are similar trends between UHII and visibility; and there is an obvious difference between trend before 2008 and trend after 2008 of both UHII and visibility. Figure 2 shows the relationships between UHII and
visibility based on their respective trends shown in Figure S2. UHII and visibility are positively correlated grossly. Higher visibility means a lower aerosol concentration, leading to a higher UHII, and vice-versa. On the other hand, the two may also change in opposite directions if the expansion of a city is more associated with the heavy industry of strong emissions. In such a case, the expansion can produce both



more aerosol particles, especially secondary aerosols converted from precursor gases, and stronger UHI,

but they have no causal relation. This is likely a reason for the diverse relationships between the trends of

the two variables. Of course, the complication originates from highly different pathways of city

expansions among these cities. The overall positive relationships revealed in Figure 2 may thus serve as

a testimony to the dominance of their causal relationship, implying that aerosol loading does influence

the UHII to a varying degree.

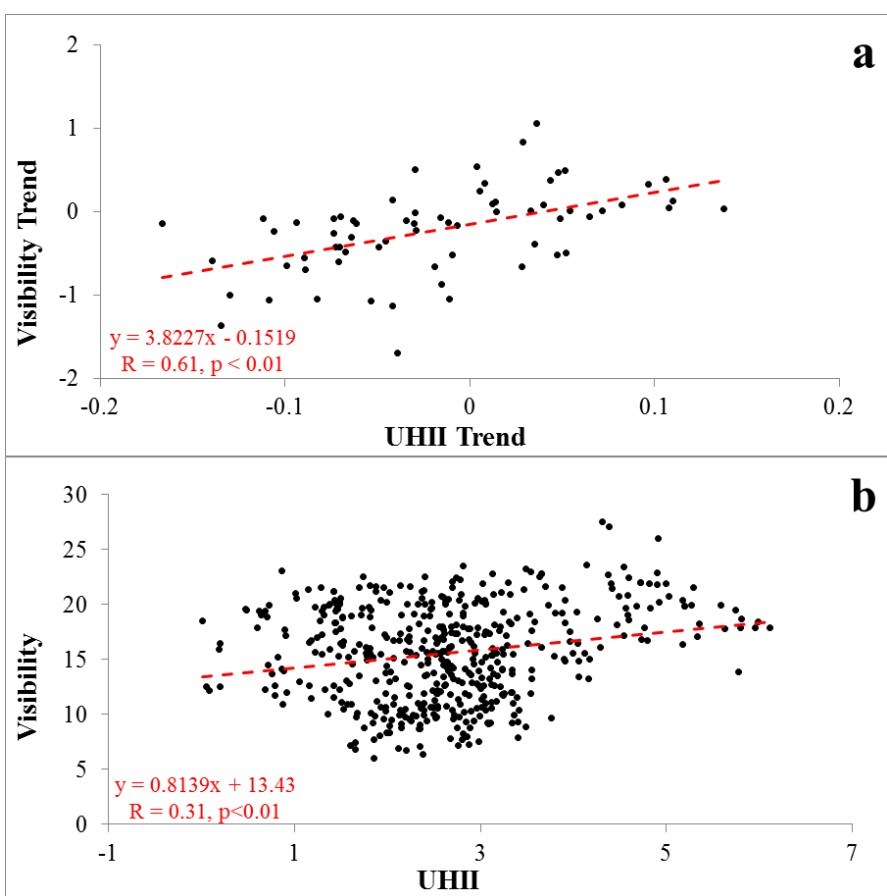


**Figure 2.** (a) Visibility trend (unit: km yr$^{-1}$) shown as a function of the UHII trend (K yr$^{-1}$), and (b) visibility (unit: km) shown as a function of UHII (unit: K). The blue line is the linear best-fit line through the points. The least-squares regression equation is given in each panel. The coefficient correlation (R) and p-value are also given.


In order to better investigate the effect of aerosols on the UHII, we calculated the UHII under severe air

pollution conditions (i.e., visibility less than 8 km) and compared it with the average UHII:





$$MUHII = \frac{\sum_{i=1}^{n=15} UHII_{m\sim i} - UHII_{p\sim i}}{n}, \qquad (3)$$

where *MUHII* is the difference between the UHII under severe air pollution conditions and the average

UHII, *n* is number of years from 2001 to 2015, *i* represents a specific year during 2001–2015, $UHII_{m\sim i}$

is the average UHII in year *i*, and $UHII_{p\sim i}$ is the UHII under severe air pollution conditions in a year *i*.

Figure S3 shows the *MUH11* at each city under polluted conditions and for all days. On an annual scale,

the UHII under severe air pollution conditions is lower than the average UHII, which suggests that a high

aerosol loading will reduce the UHII. In summer (Figure 3a), the UHI at 29 of the 35 cities is weaker

under polluted conditions. In winter (Figure 3b), however, it is just opposite, with the majority (27 out of

35) cities having stronger UHI under polluted conditions, suggesting that aerosols enhance the UHII in

winter.

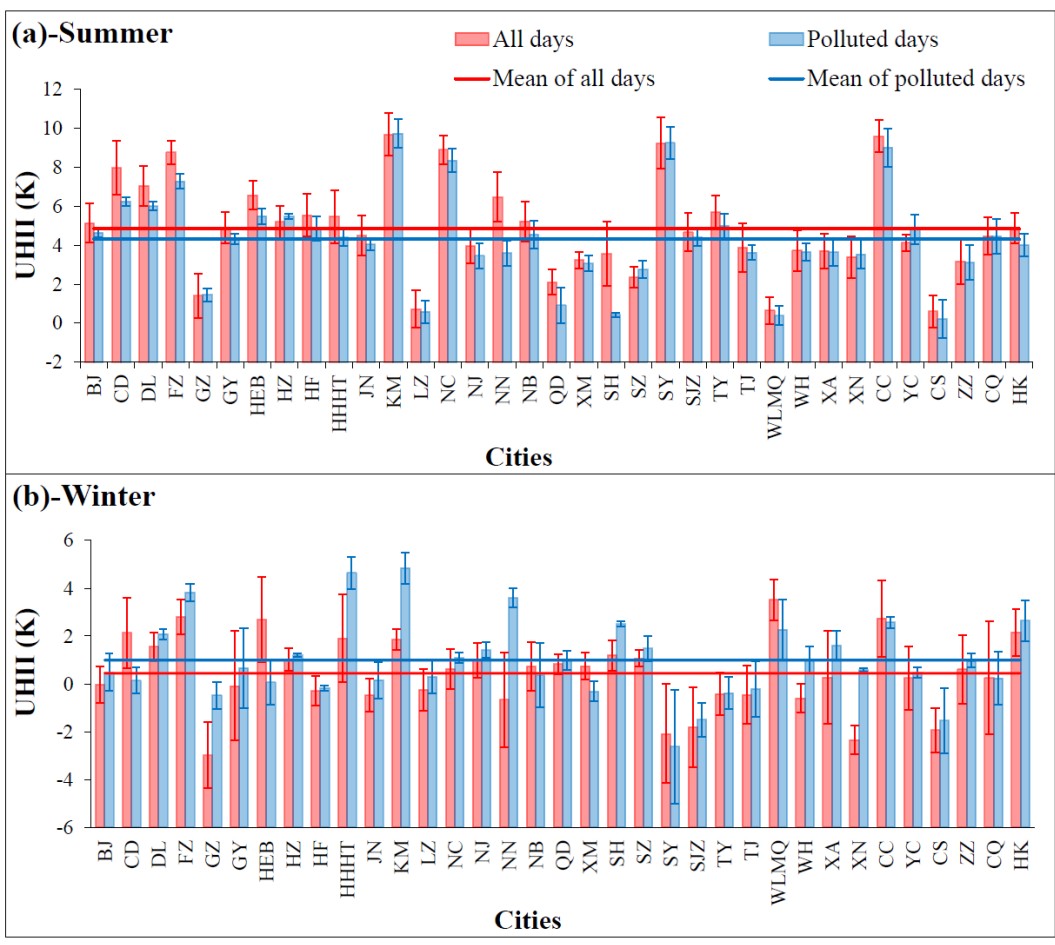





**Figure 3.** The mean UHII (unit: K) at the 35 cities in (a) summer and (b) winter. Red and blue bars represent UHII calculated using data from all days and from polluted days only, respectively. The overall mean UHII calculated using data from aS4ll days and from polluted days only are shown as red and blue solid lines, respectively.

## 4. Causes for the opposite impacts of aerosol on the UHI in summer and winter

### 4.1 Mechanisms of aerosol impact on the UHI

Aerosols alter radiation budget by scattering and absorbing solar radiation (Chylek and Coakley 1974, Chylek and Wong 1995, Li 1998). The aerosol radiative effect tends to cool down surface, warm up the atmosphere, stabilize the PBL, suppress the dispersion of pollutants in the PBL to incur a positive feedback (Li et al. 2017). As illustrated in Figure 4, the UHII may be influenced by both the Spattial Discrepancy of Aerosol Radiative Effect (SD-ARE) and the suppressed vertical exchange of surface heat fluxes which may be denoted as the Dynamic Effect (DE) for it is related to turbulent dynamics.

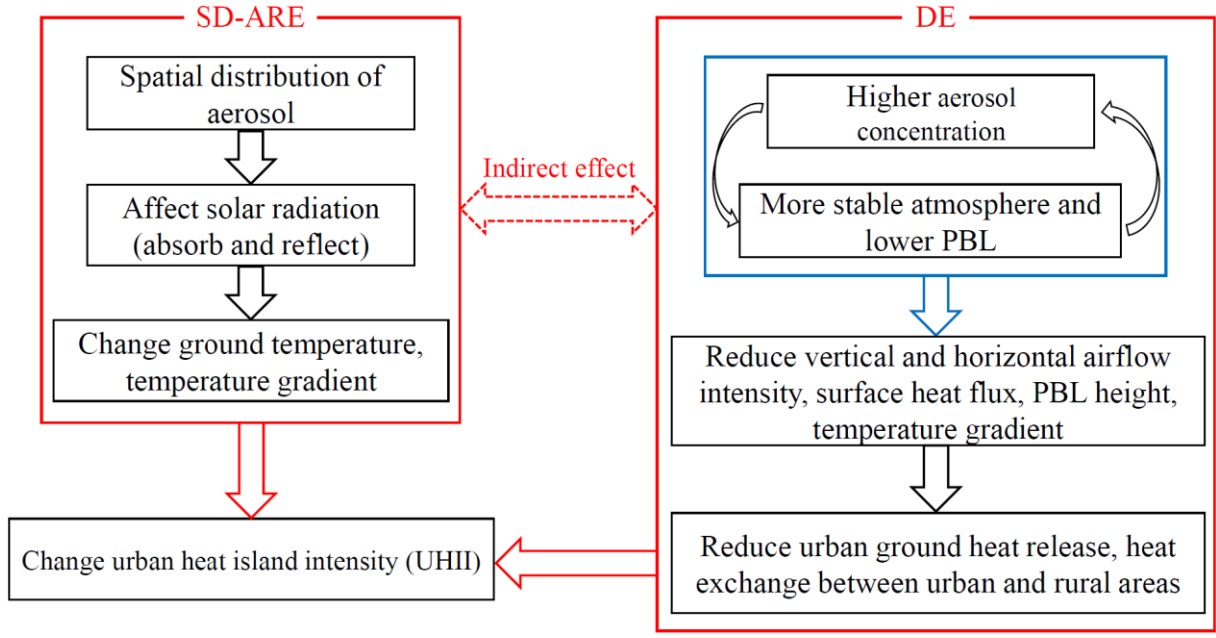

**Figure 4.** Diagram of the mechanisms behind aerosol effects on the UHII. The blue frame contains the processes and interactions between aerosols and the PBL. Red frames contain the processes of the Spatial Discrepancy of Aerosol Radiative Effect (SD-ARE) and the Dynamic Effect (DE), respectively. The solid arrows denote the direct effect, while the dashed arrow indicates the indirect effect.



*The Spatial Discrepancy of Aerosol Radiative Effect (SD-ARE):* The varying amount of solar radiation reaching the ground between urban and rural areas influences the rise of the LST and thus changes UHII,

as a result of different aerosol loading and properties between urban and rural areas. This process has a negative effect on UHII.

*The Dynamic Effect (DE):* On the other hand, aerosol-induced temperature inversion (especially in winter) within the PBL (Zhang et al. 2014, Li et al. 2015, Li et al. 2017) renders very stable PBL to inhibit vertical and horizontal airflows and surface heat fluxes (latent/sensible heat) between urban areas and rural areas

(Petäjä et al. 2016). Urban surfaces can store more heat which affects the UHII. This process has a positive effect on UHII.

Compared with rural areas, urban impervious surfaces have low thermal capacity and their temperatures are thus more sensitive to heat changes. Note that the DE and SNA-RE are not independent and that there is an indirect effect between them due to potential urban-rural circulation.


## 4.2 Analyses of Influential factors

*Urban-rural differences in air quality:* The urban-rural differences in air quality were analyzed by calculating the spatial differences of $PM_{2.5}$ and AOD under cloudless conditions between urban and rural areas. Their spatial differences are then analyzed between summer and winter:

The measurements of urban $PM_{2.5}$ concentrations were divided into four categories: 0–50, 50–100, 100–150, and > 150 μg m$^{-3}$ based on urban pollution. Figure 5 shows the mean urban-rural differences in each *PM2.5* concentration bin of all cities. On average, the spatial difference in summer is larger than in winter across all *PM2.5* concentration bins. Figure 6 shows the variation trends of mean AOD as a function of distance from the urban geometrical center of each city in winter and summer. As the distance from the

urban geometrical center increases, summertime AODs decrease more rapidly than wintertime AODs. Both Figures 5 and 6 indicate that the spatial difference of air pollution between urban and rural areas in summer is larger than that in winter. Moreover, in summer, urban pollution is often more serious than rural pollution. In winter, both urban and rural pollution is serious, and rural may be more serious than urban.





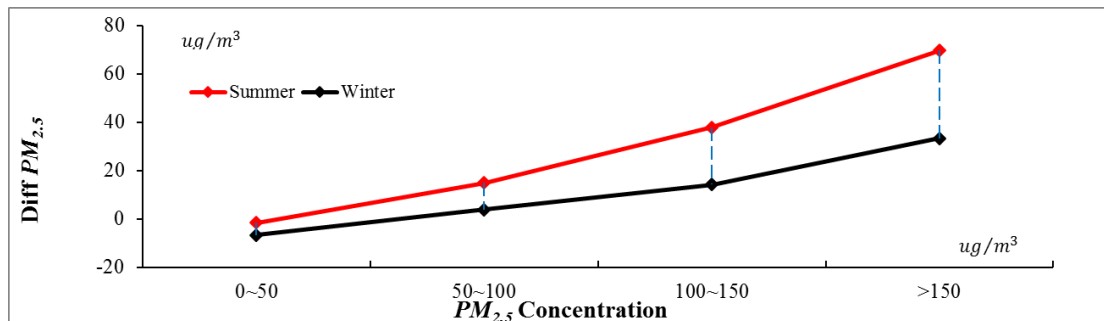


**Figure 5.** Summertime (red curves) and wintertime (black curves) urban-rural $PM_{2.5}$ (unit: µg m$^{-3}$) differences of all cities across four $PM_{2.5}$ concentration bins: 0–50, 50–100, 100–150, and > 150 µg m$^{-3}$.

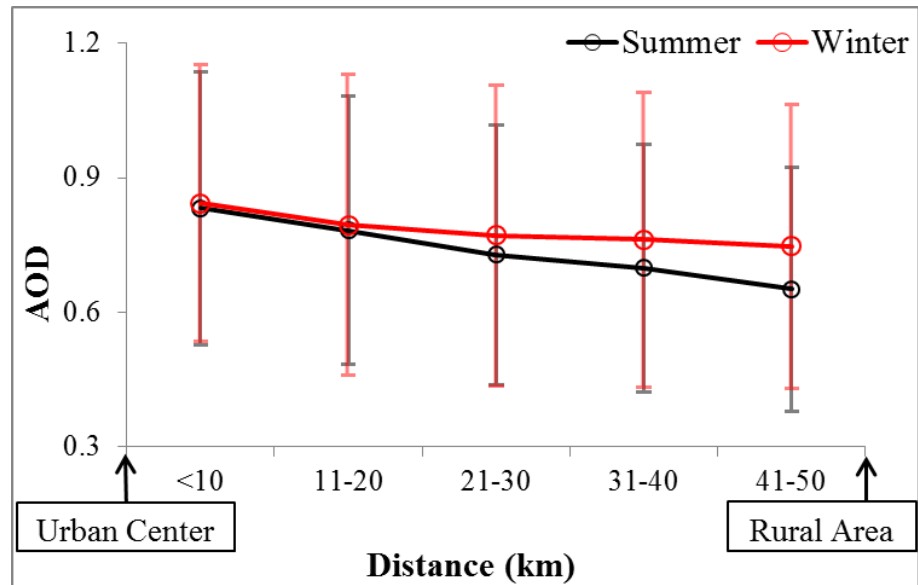

**Figure 6.** Mean AOD as a function of distance from the urban geometrical center of all cities in winter
(red curves with open circles) and summer (black curves with crosses). The distance ranges are < 10 km, 11–20 km, 21–31 km, 31–40 km, and 41–50 km from the urban geometrical center.

*Air stability inside the PBL:* Wind affects the heat exchange between urban and rural areas, regardless of wind direction, high wind speed favors urban-rural heat exchange and reduces the UHII, while low wind
speed decrease urban-rural heat exchange and enhance UHII. The means of wind speed are computed in urban and rural areas in summer and winter, under polluted and clean conditions (Figure 7). As is expected, the mean wind speed under polluted conditions is lower than that under clean conditions, especially in




winter when the difference is 1.1 ms$^{-1}$ v.s. 0.6 ms$^{-1}$ in summer. This result indicates that the urban-rural exchange in summer is stronger than that in winter.

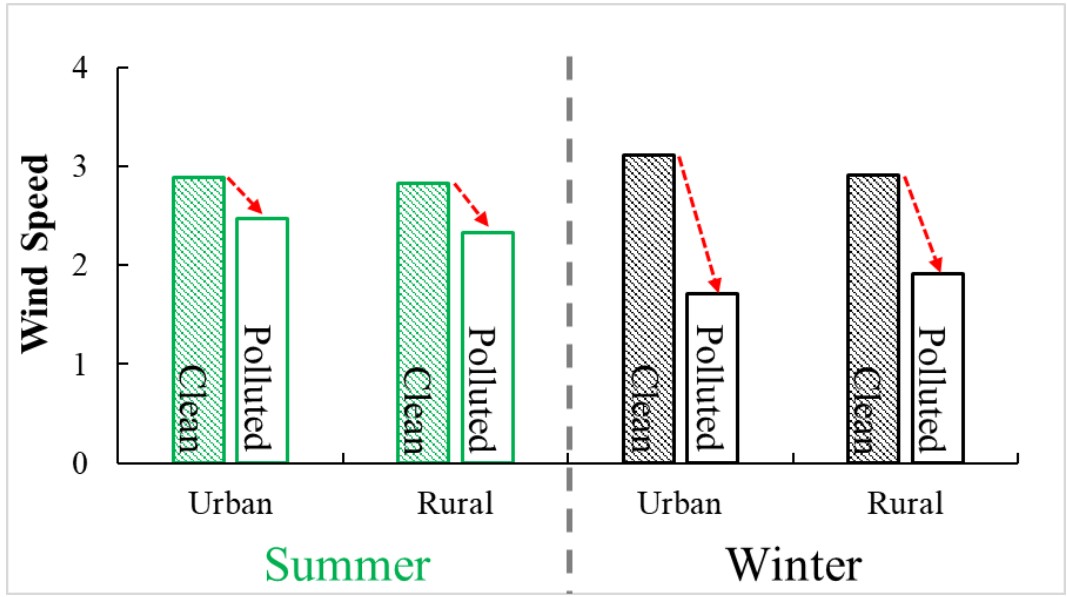


**Figure 7.** Comparison of average wind speed (unit: m s$^{-1}$) of 35 cities between urban and rural areas under heavy air pollution (white bars) and clean conditions (dark bars) in summer and winter, respectively.

Vertical temperature gradients affect the stability of the atmosphere, surface heat fluxes (especially
sensible heat) and vertical turbulence. Figure 8 shows the vertical temperature profiles at five cities in different seasons under polluted and clean conditions. Note that there are much fewer sounding stations than general surface meteorological stations in China. The vertical temperature gradient is weaker under polluted conditions than under clean conditions, so the vertical mixing is weaker. This phenomenon is also more pronounced in winter than in summer. The temperature gradient within the planetary boundary
layer (PBL) under polluted conditions generally decreases more drastically than under clean conditions, except at Chengdu that is located in the Sichuan Basin. The temperature lapse rate is smallest under polluted conditions in winter (Figure 9). These results suggest that compared with clean conditions, vertical airflow and surface heat release under polluted conditions are lessened more significantly in winter than in summer.







**Figure 8.** Mean vertical temperature profiles (vertical curves, unit ℃) at five cities in different seasons and under polluted (black) and clean (green) conditions. The mean PBL heights (PBLH) under polluted and clean conditions are also shown (black and green horizontal lines, respectively). Panels (a), (c), (e), (g), and (i) show summertime results, and panels (d), (f), (h) and (j) show wintertime results. Panels (k) and (l) show temperature reductions below 1.5 km [unit: ℃ $(100\ m)^{-1}$] under polluted (dark bars) and clean (white bars) conditions in summer and winter, respectively.



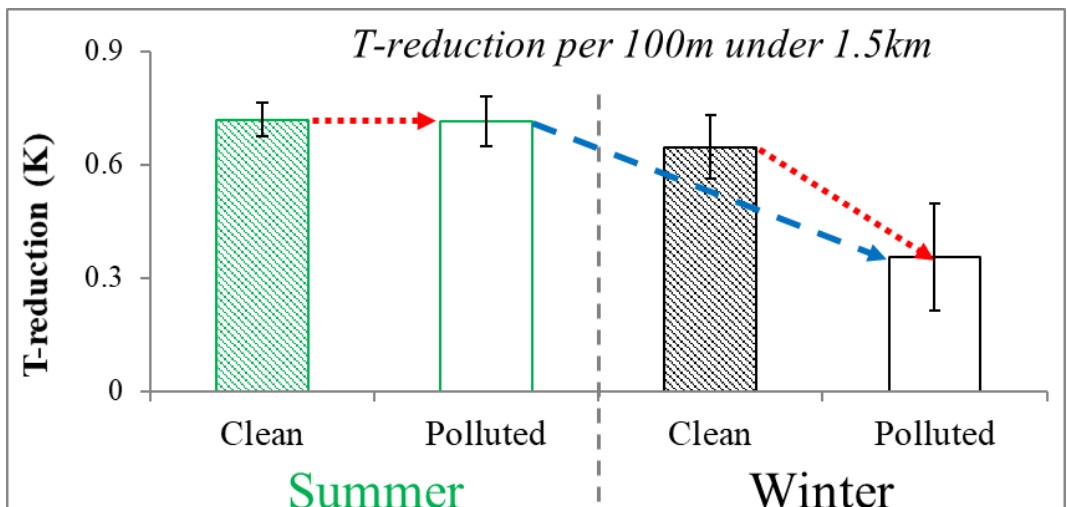

**Figure 9.** The average temperature reductions of sounding observations of five cities below 1.5 km [unit: K $(100 \text{ m})^{-1}$] under polluted (dark bars) and clean (white bars) conditions in summer and winter, respectively.

Seasonal differences in air stability in the urban and rural areas may be summarized as follows. Under polluted conditions, both horizontal and vertical exchanges decrease inside the PBL, thus weakens the heat exchange and pollution dispersion. However, this effect is much stronger in winter than in summer. In winter, airflow significantly weakens with the increasing pollution, the PBL becomes very stable, and heat exchanges significantly decrease inside the PBL.

*UHII response to variation of visibility:* Figure S5 shows the relationship between UHII and visibility difference. For most cities, higher visibility difference causes smaller UHII in summer, while UHII barely changes as visibility difference change in winter. This result indicates that UHII is more sensitive for visibility difference in summer than winter, namely, the SD-ARE has an obvious effect in summer, but it is very weak in winter.

The above analyses indicate that the two mechanisms behave differently roles in summer and winter. In summer, the SD-ARE plays a more important role than the DE to change the UHII, while the importance of two mechanisms is opposite in winter.

**4.3 Testing the mechanisms by modeling**





We evaluate the simulated aerosol and meteorological properties with surface $PM_{2.5}$ observation and sounding data (Figure S6, S7 and 10). Figure S6 shows that the simulated $PM_{2.5}$ near-surface get high concentrations at the regions south of Beijing and east to Beijing, in general agreement with the

observation. The temporal variations of simulated $PM_{2.5}$ concentrations have consistent trends as observation for most stations (Figure S7). The vertical profiles of temperature, RH and wind speeds also agree with the sound in observation. Therefore, the simulation results are sound in general.

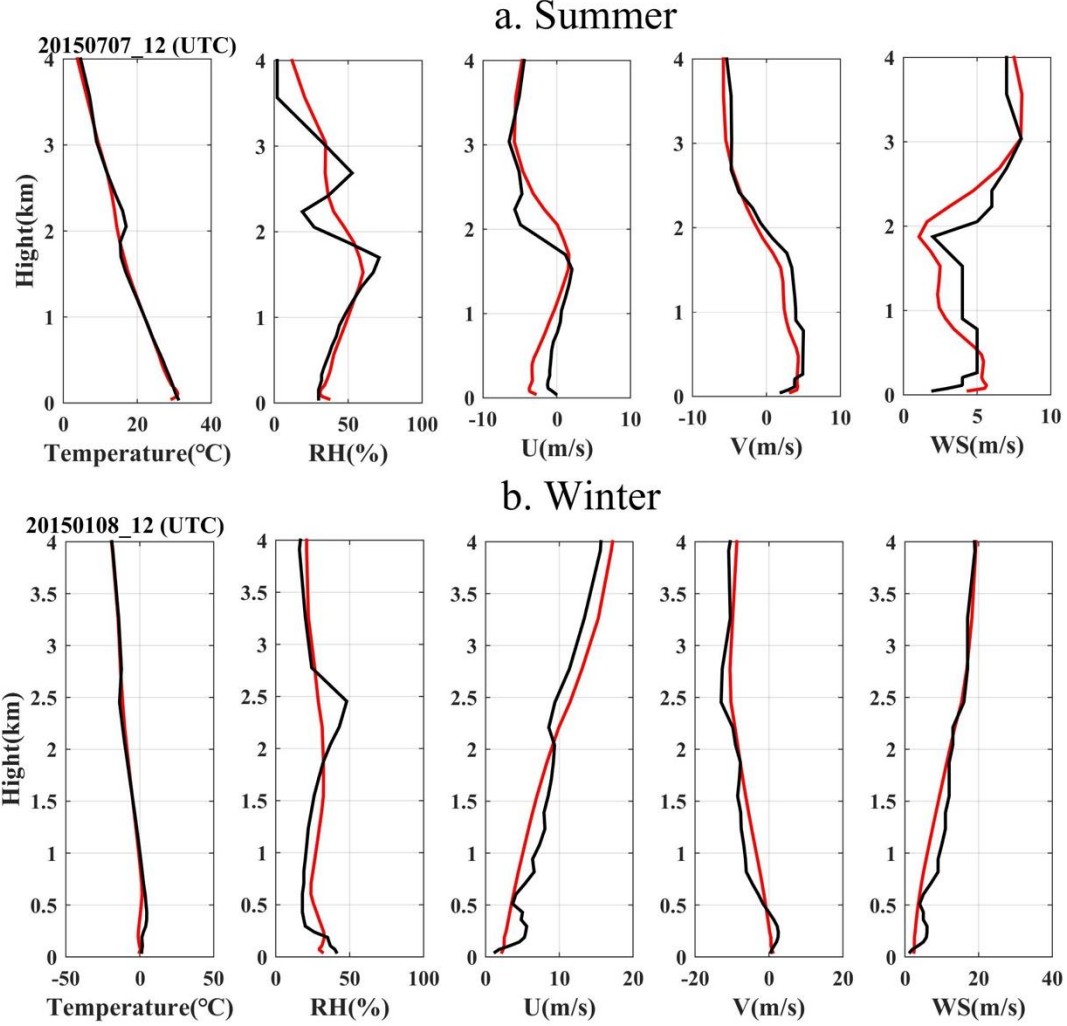

**Figure 10.** The vertical profile comparisons of temperature (unit: ℃), RH (unit: %), U-wind speed (unit:
m/s), V-wind speed (unit: m/s) and wind speed (unit: m/s) at (a) 20150708-12:00 BJT in summer and (b) 20150107-12:00 BJT in winter. The red lines are simulation results, the black lines are observation data.



Figure 11 depicts the averaged diurnal variations of UHII differences (ΔUHII) between UHII with aerosol radiation effect (ARE) and UHII without ARE, with negative values meaning the reduction of UHII by
aerosols reduce UHII and positive values showing the opposite. In summer (Figure 11 a), aerosols reduce UHII throughout all day; but in winter (Figure 11 b), aerosols enhance UHII in the afternoon. These results are consistent with the observational results shown in Figure 3. The averaged diurnal variation of downward shortwave radiation at the surface (SWDOWN) between urban and rural areas shows that the SWDOWN difference between urban and rural areas in summer is larger than that in winter (Figure S8).
The results in Figure S6 and S8 indicate that the spatial difference of air pollution in summer is larger than that in winter, and the wintertime pollution is more serious than summertime pollution, which is consistent with observational results shown in Figures 5-6. Figure S9 shows the model simulated temperature reductions below 1.5 km, suggesting the ARE is more significant on the temperature lapse rate in winter than that in summer in both urban and rural areas. Moreover, the temperature lapse rate in
summer is far more than that in winter. They are also consistent with the observational results shown in Figure 9.

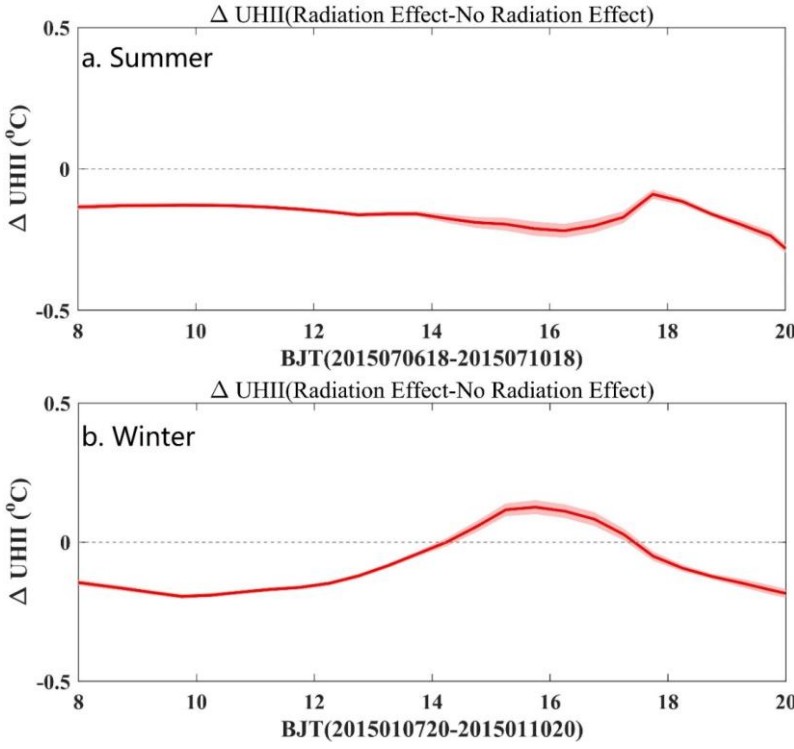


**Figure 11.** The average diurnal variation of UHII differences (Δ*UHII* unit: ℃) between UHII with ARE and UHII without ARE for typical days in (a) summer and (b) winter.


## 5. Conclusion and discussions

Satellite, ground-based, sounding data and WRF-Chem mods were used to analyze the UHII under polluted and clean conditions at 35 cities in China. Seasonal differences in UHII between summer and winter were also compared. On an annual basis, aerosols reduce the UHII, which is consistent with previous work (Wu et al. 2017). In summer, aerosols reduce the UHII, but in winter, aerosols enhance the UHII. Furthermore, we used the concepts of the Spatial Discrepancy in Aerosol Radiative Effect (SD-ARE) and the Dynamic Effect (DE) to explain how aerosols influence the UHII in different seasons. We then verified The mechanisms by means of observational analyses and model simulations.

In summer, airflow changes slightly within the PBL under polluted conditions. There is still a strong heat release and heat exchange between urban and rural areas, so the dynamic effect is weak. The spatial discrepancy in aerosol radiative effect differs between urban and rural areas because of the inhomogeneous spatial distribution of air pollution between these two areas. Since urban pollution is often more severe than rural pollution, less solar radiation reaches urban areas than in rural areas. The urban temperature enhancement is thus weaker under polluted conditions than clean ones, weakening the UHII. Figure 12a shows a diagram of how aerosols influence the UHII in summer.

In winter, the spatial discrepancy in aerosol radiative effect is weak but the dynamic effect is significant under polluted conditions in winter. Concerning the spatial discrepancy in aerosol radiative effect, the spatial difference of air pollution between urban and rural areas is small, and urban and rural areas likely experience the same severe pollution, this heats the atmosphere and reduces the similar amount of solar radiation reaching the urban and rural ground. Concerning the dynamic effect, airflow intensity and temperature gradients significantly decrease, stabilizing the PBL and weakening the heat release and heat exchange. Since pollution conditions in both urban and rural areas are similar, the spatial discrepancy in aerosol radiative effect is not a major factor causing higher UHIIs. the dynamic effect weakens airflow, reducing temperature gradients significantly, which in turn, reduces the heat exchange between urban and





rural areas and the surface heat release. Increasing heat thus accumulates in urban areas, thereby increasing the UHII. Figure 12b shows a diagram of how aerosols influence the UHII in winter.

Our analysis shows the seasonally different effects of aerosols on the UHII and explains the different mechanisms in different seasons, and the mechanism summary is shown in tables at the bottom of Figures 12a and 12b. Although this study comprehensively explains potential aerosol effects, other effects may

be at play such as land surface and aerosol properties (e.g., absorbing versus scattering aerosols). More work needs to be done to verify this. Additionally, this study analyzed observations made at 35 cities located in China and some results of a few cities that were at odds with generalized findings of 35 cities; the different results in some cities may be due to the unique characteristics of these cities regarding their location, terrain, climatic background, etc. This warrants further investigations.






**Figure 12** Aerosol effects on the UHII in (a) summer and (b) winter under polluted conditions. The background brightness indicates pollution intensity. Yellow arrows show the solar radiation, and their width indicates the amount of solar radiation to the ground. Green arrows denote heat, and their width indicates heat intensity. Black points and arrows show the process of aerosol-radiation-interaction. 420 Temperature profiles reflect the vertical temperature gradient under clean and polluted conditions. The plus sign means positive effect, while the minus sign means a negative effect.

## Data availability

The Landsat data, MODIS LST and MAIAC AOD can be download from
https://search.earthdata.nasa.gov/. Hourly PM2.5 data is published in real time by the China National
Environmental Monitoring Center (CNEMC, http://www.cnemc.cn/). National Centers for
Environmental Prediction/National Weather Service/NOAA/U.S. Department of Commerce (2000):
NCEP FNL Operational Model Global Tropospheric Analyses, continuing from July 1999. Research Data
Archive at the National Center for Atmospheric Research, Computational and Information Systems
Laboratory. Dataset. Accessed 22 Feb. 2019. https://doi.org/10.5065/D6M043C6.

## Author contributions

All authors made substantial contributions to this work. HWC and LZQ designed this research. HWC
conducted the analyses and wrote the draft under the supervision of LZQ. LZQ reviewed and edited this
paper. WF and ZYW conducted the WRF-Chem simulation and helped to edit this paper. GJP and STN
provided part of datasets and helped to edit this paper. MC reviewed and edited this paper. CTM and WJ
provided part of datasets. LSS gave many suggestions about model simulations.

## Competing interests

The authors declare that they have no conflict of interest.

## Acknowledgements

This study has been supported by the National Key R&D Program of China (2017YFC1501702), the
National Nature Science Foundation of China (91544217 and 41771399), the U.S. National Science
Foundation (AGS1534670) and the U.S. Department of Energy Office of Science Early Career Award
Program.

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
