# Peer review of "The Mechanisms and Seasonal Differences of the Impact of Aerosols on Daytime Surface Urban Heat Island Effect"

_Atmospheric Chemistry and Physics, 2020_

## Referee Comment (RC1) · Anonymous Referee #1 · 28 Feb 2020

General comments: This study showed that that aerosols have very different effects on daytime UHII in different seasons: reducing the UHII in summer, but increasing the UHII in winter. It also found that he seasonal contrast in the spatial distribution of aerosols between the urban centers and the suburbs lead to a spatial discrepancy in aerosol radiative effect. Different mechanisms are analyzed for different seasons. The manuscript has well-presented some interesting findings. However, there are still some major concerns that need to be addressed.

Some concerns: 1.I suggest the authors combine the first paragraph with the second paragraph. And remove the sentence in the second paragraph, "It is well established

that cities are the largest sources of anthropogenic heat emissions as by-products from industrial and human activities. Human activities can also generate large amounts of aerosols that can reduce air quality, change the physical and chemical properties of the atmosphere, and endanger human health (Sanap and Pandithurai 2015, Cohen et al. 2017, Wei et al. 2019a, b)." to the beginning of the third paragraph. 2.The sentence "The effect of urbanization on clouds and precipitation has been the focus of many studies (Changnon et al. 1977, Ackerman et al. 1978, Changnon et al. 1991, Shepherd et al. 2002, Shepherd and Burian 2003)" looks very abrupt here. 3. Overall, illustrations with more logics are needed for introduction Section. 4.Section 2 is suggested to be "Data and methods". The data, method of Extracting urban impervious surfaces and urban contours, model description should be included separately by three parts. I suggest the authors combine the "research windows" into "method of Extracting urban impervious surfaces and urban contours". And combine the "Aerosol parameters" to the part of "data" together with some data in the present part "Study areas and data". "Study areas" can be removed to the beginning of result analysis. Additionally, the method of calculating USII should be mentioned. 5. Figure 1 need some modifications, for example, the city name "Changchun" has been divided into 2 rows. 6. Figure 2 should mark the result of confidence test. The impact factors are more than aerosol. The atmospheric humidity should be included. Here, the relation between USII and visibility may be one-sided. Moreover, the result of urban heat island does not just include the changes in aerosol. In Figure 3, the details of "severe air pollution condition" should be shown, including the data used here, definition of severe air pollution and period of severe air pollution event. It is not clear whether the ðIŚĹðIŘżðIŘijðIŘijðIŠÍ~ðIŚŰ is calculated based on several severe air pollution event or not. The details should be addressed and added. 7. Figure 5, "red curves" should be changed to "red curve", "black curves" to "black curve". How did the PM2.5 concentration bins get? And was the urban-rural PM2.5 difference corresponded to the PM2.5 concentration bins? Some descriptions should be added. Same information for AOD is also needed. 8.A better quality of combination figure is needed for Fig. 8. The meaning of arrows in Fig. 9
should be given. 9.Page 19, Line 368, "In summer (Figure 11 a), aerosols reduce UHII throughout all day; but in winter (Figure 11 b), aerosols enhance UHII in the afternoon. These results are consistent with the observational results shown in Figure 3". Figure 3 cannot provide the consistent result. Figure 3 shows the result at annual scale. However, Figure 11 is the simulation for an event for three days.

---

## Referee Comment (RC2) · Anonymous Referee #2 · 4 Mar 2020

This study investigated the relationships between daytime surface urban-heat-island (SUHI) intensity and aerosol pollution in summer and winter and their seasonal difference in China by using multi-source observations. The topic is very interesting and has important climate, environment and health implications. This study has the potential to provide new insights about urban climate change and their seasonal change under heavy air pollution context. The manuscript is written clearly, and I really like the schematic diagram in figure 12. While I found some minor issues need to be addressed. My recommendation is to accept with minor revision. 1. Introduction: UHI can be defined by satellite-based Land surface temperature (LST) (i.e., surface UHI) and also can be defined by surface air temperature (SAT) recorded by stations. There

still is a bit differences in these two definitions and their drive factors, although SAT is closely related to LST. Therefore, some papers in the Introduction need to be stated clearly for which definition. In addition, suggest that surface is added in the paper Title and the MS. 2. Method: about meteorological station should be added in figure 1 or in the supplementary? How did you choose the urban and rural (i.e. reference) stations in each city? 3. Sample numbers should be added in Figure 2. 4. Please check whether eq.3 matched with the text in lines 239-240? Also y-axis title should be MUHII (or MSUHII) ? 5. Lines 251: aS4ll? What do you mean? 6. Lines 272-276: excepting temperature inversion-induced stable PBL, aerosol pollutions usually accompanied with low wind speed (particularly <2m/s), which is also favorable to both heat accumulation/store and UHI enhancement. 7. Lines 292-293: In the daytime during winter, the high aerosol concentrations in the rural areas, due to high emission induced by coal heating in rural area in the north China, while in south more industries in rural areas under stagnant atmospheric conditions? This seasonal variation in urban-rural difference may modulated by the combined effects of PM2.5 emission, transportation and diffusion (please refer to Urban-rural differences in PM2.5 concentrations in the representative cities of China during 2015~2018. CHINA ENVIRONMENTAL SCIENCECE, 2019, 39(11): 4552-4560.). 8. Subfigures in Figure S5 and S7 are very small unclear for readership.

---

## Short Comment (SC1) · 15 Apr 2020

General comments: This study investigated the relationships between daytime surface urban-heat-island (SUHI) intensity and aerosol pollution in summer and winter and their seasonal difference in China by using multi-source observations. The topic is very interesting and has important climate, environment and health implications. This study has the potential to provide new insights about urban climate change and their seasonal change under heavy air pollution context. The manuscript is written clearly, and I really like the schematic diagram in figure 12. While I found some minor issues need to be addressed. My recommendation is to accept with minor revision. We would

like to thank you very much for providing so many insightful comments. Following your comments and suggestions, we have made many changes. The manuscript was also more carefully edited. The responses are highlighted in red; the changes in manuscript are highlighted in blue in this response file.

Comment 1. Introduction: UHI can be defined by satellite-based Land surface temperature (LST) (i.e., surface UHI) and also can be defined by surface air temperature (SAT) recorded by stations. There still is a bit differences in these two definitions and their drive factors, although SAT is closely related to LST. Therefore, some papers in the Introduction need to be stated clearly for which definition. In addition, suggest that surface is added in the paper Title and the MS. Response: We agree with you that UHII mainly contains surface and atmospheric urban heat islands. We added the sentence "While the UHI mainly involves surface and atmospheric UHIs, this study focuses on surface UHI." on line 57. We also changed the title to "The Mechanisms and Seasonal Differences of the Impact of Aerosols on Daytime Surface Urban Heat Island Effect" in the manuscript and supplement.

Comment 2. Method: about meteorological station should be added in figure 1 or in the supplementary? How did you choose the urban and rural (i.e. reference) stations in each city? Response: We added the spatial distribution of meteorological stations in Figure S1:

Figure S1. Spatial distribution of meteorological stations located in 35 cities.

We added the sentence "Figure S1 shows the spatial distribution of the meteorological stations." on line 138. We selected these stations city by city. Urban stations are those stations located within the urban boundaries shown in Figure 2. Rural stations are those stations located outside urban boundaries, least affected by urban areas and with the lowest altitude difference with the urban areas.

Comment 3. Sample numbers should be added in Figure 2. Response: Figure 3 results are based on Figure S5. Samples numbers in Figure 3a and Figure 3b are 68

and 510, respectively (see line 250).

Comment 4.   Please check whether eq.3 matched with the text in lines 239-240?   Also y-axis title should be MUHII (or MSUHII)? Response:   In Figures 4 and S4, we directly compared UHII under polluted conditions and UHII for all days.   We deleted the previous version of the figure.   The following passage has been deleted:   MUHII= $(\sum_{(} i = 1)^{(} n = 15) UHII_{(m\ i)} - UHII_{(p\ i)})/n, (3) where MUHII is the difference between the UHII under severe air pollution conditions and the average UHII ... -2015, UHII_{(m\ i)} is the average UHII in year i, and UHII_{(p\ i)} is the UHII under severe air pollution conditions in a year i. Figur$

Comment 5. Lines 251: aS4ll? What do you mean? Response: Here, "sS4ll" should be "all". We have corrected this typo.

Comment 6.   Lines 272-276: excepting temperature inversion-induced stable PBL, aerosol pollutions usually accompanied with low wind speed (particularly <2m/s), which is also favorable to both heat accumulation / store and UHI enhancement. Response: Yes, you are right. We added more explanations on lines 289 to 291: "In addition to a temperature-inversion-induced stable PBL, air pollution is usually accompanied by low wind speeds (particularly < 2m s-1), also favorable to both heat accumulation and storage."

Comment 7. Lines 292-293: In the daytime during winter, the high aerosol concentrations in the rural areas, due to high emission induced by coal heating in rural area in the north China, while in south more industries in rural areas under stagnant atmospheric conditions? This seasonal variation in urban-rural difference may modulated by the combined effects of PM2.5 emission, transportation and diffusion (please refer to Urban-rural differences in PM2.5 concentrations in the representative cities of China during 20152018. CHINA ENVIRONMENTAL SCIENCECE, 2019, 39(11): 4552-4560.). Response: We agree. We added the following sentence (lines 313 to 314): "Many factors (e.g., PM2.5 emissions, transportation, and diffusion) may cause the seasonal difference in urban-rural differences (Jiang et al., 2019)."

[Figure]

We also added the reference (line 560): "Jiang, Y. C., Yang, Y. J., Wang, H., Li, Y. B., Gao, Z. Q., and Zhao, C.: Urban-rural differences in PM2.5 concentrations in the representative cities of China during 2015–2018, China Environ. Sci., 39(11), 4552–4560, https://doi.org/10.19674/j.cnki.issn1000-6923.2019.0530, 2019."

Note that we have another manuscript under review, focused on the seasonal difference in the urban-rural spatial distribution of air pollution, which also analyzes several potential factors that cause the seasonal difference.

Comment 8. Subfigures in Figure S5 and S7 are very small unclear for readership. Response: Figures S6 and S8 have been redrawn for clarity. Increasing the zoom percentage will help with seeing more details.

Please also note the supplement to this comment:
https://www.atmos-chem-phys-discuss.net/acp-2020-162/acp-2020-162-SC1-supplement.pdf
* * *
Urban stations

Rural stations

High : 8463

Low : -155

**Fig. 1.** Figure S1. Spatial distribution of meteorological stations located in 35 cities.

---

## Short Comment (SC2) · 15 Apr 2020

To Anonymous Referee 1 General comments: This study showed that that aerosols have very different effects on daytime UHII in different seasons: reducing the UHII in summer, but increasing the UHII in winter. It also found that he seasonal contrast in the spatial distribution of aerosols between the urban centers and the suburbs lead to a spatial discrepancy in aerosol radiative effect. Different mechanisms are analyzed for different seasons. The manuscript has well-presented some interesting findings. However, there are still some major concerns that need to be addressed. Response: We would like to thank you very much for providing so many insightful comments. Following your comments and suggestions, we have made many changes. The manuscript was also more carefully edited. The responses are highlighted in red; the changes in manuscript are highlighted in blue in this response file.

Comment 1.I suggest the authors combine the first paragraph with the second paragraph. And remove the sentence in the second paragraph, "It is well established that cities are the largest sources of anthropogenic heat emissions as by-products from industrial and human activities. Human activities can also generate large amounts of aerosols that can reduce air quality, change the physical and chemical properties of the atmosphere, and endanger human health (Sanap and Pandithurai 2015, Cohen et al. 2017, Wei et al. 2019a, b)." to the beginning of the third paragraph. Response: Done per the suggestion.

Comment 2.The sentence "The effect of urbanization on clouds and precipitation has been the focus of many studies (Changnon et al. 1977, Ackerman et al. 1978, Changnon et al. 1991, Shepherd et al. 2002, Shepherd and Burian 2003)" looks very abrupt here. Response: We have moved this sentence to line 79 and revised this sentence to "The effect of urbanization on clouds and precipitation has also been the focus of many studies (Changnon et al., 1977; Ackerman et al., 1978; Changnon et al., 1991; Shepherd et al., 2002; Shepherd and Burian, 2003)".

Comment 3.Overall, illustrations with more logics are needed for introduction Section. Response: Thank you for this suggestion. Figure 1 is a newly added figure showing the relationship between urbanization, urban heat island, aerosols, local and regional weather and climate.

Figure 1. Illustration of the relationship between urbanization, urban heat island, aerosols, local and regional weather and climate. Solid arrows denote the effect.

Moreover, to be more logical, we moved the sentence "With increasing urbanization in the future, cities are likely to influence local and regional weather and climate to greater and greater degrees." to line 82.

Comment 4.Section 2 is suggested to be "Data and methods". The data, method of Extracting urban impervious surfaces and urban contours, model description should be included separately by three parts. I suggest the authors combine the "research windows" into "method of Extracting urban impervious surfaces and urban contours". And combine the "Aerosol parameters" to the part of "data" together with some data in the present part "Study areas and data". "Study areas" can be removed to the beginning of result analysis. Additionally, the method of calculating USII should be mentioned. Response: Done per your suggestion. The method of calculating UHII is added to section 2.2 on lines 190 to 194: "The UHII is the temperature difference between the average temperature of the urban core window and the average temperature of rural windows, calculated as UHII=$\Delta$T=T_u-T_r , (3) where T_u is the average temperature of an urban area, and T_r is the average temperature of the neighboring rural area."

Comment 5.Figure 1 need some modifications, for example, the city name "Changchun" has been divided into 2 rows. Response: We have redrawn this figure.

Comment 6.Figure 2 should mark the result of confidence test. The impact factors are more than aerosol. The atmospheric humidity should be included. Here, the relation between USII and visibility may be one-sided. Moreover, the result of urban heat island does not just include the changes in aerosol. In Figure 3, the details of "severe air pollution condition" should be shown, including the data used here, definition of severe air pollution and period of severe air pollution event. It is not clear whether the MUHII= $(\sum\_(i = 1〖n = 15)UHII\_(m\sim i)$-ãĂŰUHIIãĂŮ\_(p\sim i) ãĂŮ)/n is calculated based on several severe air pollution event or not. The details should be addressed and added. Response: We did the confidence test (95%) and added this information to Figure 3 on line 252. We agree with you that aerosols are not the only factor affecting UHII, so we added the relationship between UHII and relative humidity (RH) to Figure 3. Figure 3b shows that there is a positive correlation between RH and UHII, but it is less significant than the correlation between UHII and visibility (p-value of visibility > RH).

Figure 3. (a) Clear-day visibility trend (unit: km yr-1) shown as a function of the UHII

trend (K yr-1), and (b) clear-day visibility (unit: km) and relative humidity shown as a function of UHII (unit: K). The period is 2001-2015. The black and green lines is the linear best-fit line through the points. Sample numbers of (a) and (b) are 68 and 510 respectively. The least-squares regression equation is given in each panel. The coefficient correlation (R) and p-value are also given, and all of them pass confidence test in 95%.

The new figure updates the previous figure, and the following sentence was added (lines 242 to 246): "Also analyzed was the relationship between RH and UHII. Figure 3b shows that there is a positive correlation between RH and UHII, but it is less significant than the correlation between UHII and visibility (p-value of visibility > RH). Note that not only these two factors affect UHII. Many other factors affect UHII, but this study mainly focuses on the aerosol effect."

We have added information about "polluted days" in the upper right corner of Figure 4a. In Figures 4 and S4, we directly compared UHII under polluted conditions and UHII for all days. We deleted the previous version of the figure and the following sentence: MUHII= $(\sum\_(i = 1⊕n = 15)UHII\_(m{\sim}i)$-ãĂŰUHIIãĂŮ\_(p${\sim}$i) ãĂŮ)/n, (3) where MUHII is the difference between the UHII under severe air pollution conditions and the average UHII, n is number of years from 2001 to 2015, i represents a specific year during 2001–2015, ãĂŰUHIIãĂŮ\_(m${\sim}$i) is the average UHII in year i, and ãĂŰUHIIãĂŮ\_(p${\sim}$i) is the UHII under severe air pollution conditions in a year i. Figure S3 shows the MUH11 at each city under polluted conditions and for all days.

Comment 7.Figure 5, "red curves" should be changed to "red curve", "black curves" to "black curve". How did the PM2.5 concentration bins get? And was the urban-rural PM2.5 difference corresponded to the PM2.5 concentration bins? Some descriptions should be added. Same information for AOD is also needed. Response: Done per your suggestion. Here we divided PM2.5 concentration into four bins, and calculated the PM2.5 concentration difference in each bin, the method was shown in lines 302 to 303. Therefore, the PM2.5 difference is corresponded to each PM2.5 concentration

bins. We also added more information about AOD (lines 305 to 307): "Five zones were selected based on the distance to the urban geometric center of all cities: Zone 1: 0–10 km, Zone 2: 11–20 km, Zone 3: 21–30 km, Zone 4: 31–40 km, and Zone 5: 41–50 km, then the average AOD for each zone was calculated."

Comment 8. A better quality of combination figure is needed for Fig. 8. The meaning of arrows in Fig. 9 should be given. Response: We have redrawn Figure 9. Regarding Figure 10 (lines 365 to 367): "The red arrows mean the change ranges from clean conditions to polluted conditions, the blue arrow mean the change range from summer-polluted condition to winter-polluted condition."

Comment 9. Page 19, Line 368, "In summer (Figure 11 a), aerosols reduce UHII throughout all day; but in winter (Figure 11 b), aerosols enhance UHII in the afternoon. These results are consistent with the observational results shown in Figure 3". Figure 3 cannot provide the consistent result. Figure 3 shows the result at annual scale. However, Figure 11 is the simulation for an event for three days. Response: Figure 4 shows that aerosols reduce UHII in summer and enhance UHII in winter. Figure 12 shows similar results on a daily scale. To express this more accurately, we changed the sentence "These results are consistent with the observational results shown in Figure 4" to "This shows the effect of aerosols on UHII on a daily scale, supporting Fig. 4." on line 394.

Please also note the supplement to this comment:
https://www.atmos-chem-phys-discuss.net/acp-2020-162/acp-2020-162-SC2-supplement.pdf
* * *
[Figure]

**Fig. 1.** Figure 1. Illustration of the relationship between urbanization, urban heat island, aerosols, local and regional weather and climate. Solid arrows denote the effect.

[Figure]

a

y = 3.8227x - 0.1519
R = 0.61, p < 0.01

b

y = 0.0215x + 0.4681
R = 0.21, p<0.05

y = 0.8139x + 13.43
R = 0.31, p<0.01

**Fig. 2.** Figure 3. (a) Clear-day visibility trend (unit: km yr-1) shown as a function of the UHII trend (K yr-1), and (b) clear-day visibility (unit: km) and relative humidity shown as a function of UHII (unit

---

## Author Comment (AC2) · 17 Apr 2020

General comments: This study investigated the relationships between daytime surface urban-heat-island (SUHI) intensity and aerosol pollution in summer and winter and their seasonal difference in China by using multi-source observations. The topic is very interesting and has important climate, environment and health implications. This study has the potential to provide new insights about urban climate change and their seasonal change under heavy air pollution context. The manuscript is written clearly, and I really like the schematic diagram in figure 12. While I found some minor issues need to be addressed. My recommendation is to accept with minor revision.

[Figure]

We would like to thank you very much for providing so many insightful comments. Following your comments and suggestions, we have made many changes. The manuscript was also more carefully edited. The responses are highlighted in red; the changes in manuscript are highlighted in blue in this response file.

Comment 1. Introduction: UHI can be defined by satellite-based Land surface temperature (LST) (i.e., surface UHI) and also can be defined by surface air temperature (SAT) recorded by stations. There still is a bit differences in these two definitions and their drive factors, although SAT is closely related to LST. Therefore, some papers in the Introduction need to be stated clearly for which definition. In addition, suggest that surface is added in the paper Title and the MS.

Response: We agree with you that UHII mainly contains surface and atmospheric urban heat islands. We added the sentence "While the UHI mainly involves surface and atmospheric UHIs, this study focuses on surface UHI." on line 57. We also changed the title to "The Mechanisms and Seasonal Differences of the Impact of Aerosols on Daytime Surface Urban Heat Island Effect" in the manuscript and supplement.

Comment 2. Method: about meteorological station should be added in figure 1 or in the supplementary? How did you choose the urban and rural (i.e. reference) stations in each city?

Response: We added the spatial distribution of meteorological stations in Figure S1: We added the sentence "Figure S1 shows the spatial distribution of the meteorological stations." on line 138. We selected these stations city by city. Urban stations are those stations located within the urban boundaries shown in Figure 2. Rural stations are those stations located outside urban boundaries, least affected by urban areas and with the lowest altitude difference with the urban areas.

Comment 3. Sample numbers should be added in Figure 2.

Response: Figure 3 results are based on Figure S5. Samples numbers in Figure 3a

Interactive
comment

and Figure 3b are 68 and 510, respectively (see line 250).

Comment 4. Please check whether eq.3 matched with the text in lines 239-240? Also y-axis title should be MUHII (or MSUHII)?

Response: In Figures 4 and S4, we directly compared UHII under polluted conditions and UHII for all days. We deleted the previous version of the figure. The following passage has been deleted: MUHII= $(\sum\_(i = 1)\Theta n = 15)UHII\_(m\sim i)$-ãĂŰUHIIãĂŮ\_(p$\sim$i) ãĂŮ)/n, (3) where MUHII is the difference between the UHII under severe air pollution conditions and the average UHII, n is number of years from 2001 to 2015, i represents a specific year during 2001–2015, ãĂŰUHIIãĂŮ\_(m$\sim$i) is the average UHII in year i, and ãĂŰUHIIãĂŮ\_(p$\sim$i) is the UHII under severe air pollution conditions in a year i. Figure S3 shows the MUH11 at each city under polluted conditions and for all days.

Comment 5. Lines 251: aS4ll? What do you mean?

Response: Here, "sS4ll" should be "all". We have corrected this typo.

Comment 6. Lines 272-276: excepting temperature inversion-induced stable PBL, aerosol pollutions usually accompanied with low wind speed (particularly <2m/s), which is also favorable to both heat accumulation / store and UHI enhancement.

Response: Yes, you are right. We added more explanations on lines 289 to 291: "In addition to a temperature-inversion-induced stable PBL, air pollution is usually accompanied by low wind speeds (particularly < 2m s-1), also favorable to both heat accumulation and storage."

Comment 7. Lines 292-293: In the daytime during winter, the high aerosol concentrations in the rural areas, due to high emission induced by coal heating in rural area in the north China, while in south more industries in rural areas under stagnant atmospheric conditions? This seasonal variation in urban-rural difference may modulated by the combined effects of PM2.5 emission, transportation and diffusion (please refer to Urban-rural differences in PM2.5 concentrations in the representative cities

of China during 20152018. CHINA ENVIRONMENTAL SCIENCECE, 2019, 39(11): 4552-4560.).

Response: We agree. We added the following sentence (lines 313 to 314): "Many factors (e.g., PM2.5 emissions, transportation, and diffusion) may cause the seasonal difference in urban-rural differences (Jiang et al., 2019)." We also added the reference (line 560): "Jiang, Y. C., Yang, Y. J., Wang, H., Li, Y. B., Gao, Z. Q., and Zhao, C.: Urban-rural differences in PM2.5 concentrations in the representative cities of China during 2015–2018, China Environ. Sci., 39(11), 4552–4560, https://doi.org/10.19674/j.cnki.issn1000-6923.2019.0530, 2019." Note that we have another manuscript under review, focused on the seasonal difference in the urban-rural spatial distribution of air pollution, which also analyzes several potential factors that cause the seasonal difference.

Comment 8. Subfigures in Figure S5 and S7 are very small unclear for readership.

Response: Figures S6 and S8 have been redrawn for clarity. Increasing the zoom percentage will help with seeing more details.
* * *
[Figure]

**Fig. 1.** Figure S1. Spatial distribution of meteorological stations located in 35 cities.